# Biochemical and Phylogenetic Analysis of Italian *Phaseolus vulgaris* Cultivars as Sources of α-Amylase and α-Glucosidase Inhibitors

**DOI:** 10.3390/plants12162918

**Published:** 2023-08-11

**Authors:** Stefania Peddio, Sonia Lorrai, Alessandra Padiglia, Faustina B. Cannea, Tinuccia Dettori, Viviana Cristiglio, Luigi Genovese, Paolo Zucca, Antonio Rescigno

**Affiliations:** 1Department of Biomedical Sciences (DiSB), University Campus, Monserrato, 09042 Cagliari, Italy; s.peddio@unica.it (S.P.); s.lorrai7@studenti.unica.it (S.L.); dettorit@unica.it (T.D.); rescigno@unica.it (A.R.); 2Department of Life and Environmental Sciences (DiSVA), University Campus, Monserrato, 09042 Cagliari, Italy; padiglia@unica.it (A.P.); faustinabarbara@tiscali.it (F.B.C.); 3Institut Laue-Langevin, 38042 Grenoble, France; cristiglio@ill.eu; 4CEA/MEM/L-Sim, University Grenoble Alpes, 38044 Grenoble, France; luigi.genovese@cea.fr

**Keywords:** amylase, low-carb diet, biodiversity, bean, diabetes, phytohemagglutinin

## Abstract

*Phaseolus vulgaris* α-amylase inhibitor (α-AI) is a protein that has recently gained commercial interest, as it inhibits mammalian α-amylase activity, reducing the absorption of dietary carbohydrates. Numerous studies have reported the efficacy of preparations based on this protein on the control of glycaemic peaks in type-2 diabetes patients and in overweight subjects. A positive influence on microbiota regulation has also been described. In this work, ten insufficiently studied Italian *P. vulgaris* cultivars were screened for α-amylase- and α-glucosidase-inhibiting activity, as well as for the absence of antinutritional compounds, such as phytohemagglutinin (PHA). All the cultivars presented α-glucosidase-inhibitor activity, while α-AI was missing in two of them. Only the Nieddone cultivar (ACC177) had no haemagglutination activity. In addition, the partial nucleotide sequence of the α-AI gene was identified with the degenerate hybrid oligonucleotide primer (CODEHOP) strategy to identify genetic variability, possibly linked to functional α-AI differences, expression of the α-AI gene, and phylogenetic relationships. Molecular studies showed that α-AI was expressed in all the cultivars, and a close similarity between the Pisu Grogu and Fasolu cultivars’ α-AI and α-AI-4 isoform emerged from the comparison of the partially reconstructed primary structures. Moreover, mechanistic models revealed the interaction network that connects α-AI with the α-amylase enzyme characterized by two interaction hotspots (Asp38 and Tyr186), providing some insights for the analysis of the α-AI primary structure from the different cultivars, particularly regarding the structure–activity relationship. This study can broaden the knowledge about this class of proteins, fuelling the valorisation of Italian agronomic biodiversity through the development of commercial preparations from legume cultivars.

## 1. Introduction

Over 1300 species of legumes are known in the world, but only 20 are commonly consumed by humans. The common bean, *Phaseolus vulgaris*, is the grain legume most commonly eaten around the globe [1], representing an important source of dietary protein (constituting 22–25% of seed weight) for billions of people and livestock [2,3].

In recent decades, common beans have also been considered nutraceutical foods due to their content of inhibitors of carbohydrate-hydrolysing enzymes, such as proteinaceous α-amylase inhibitor (α-AI) and α-glucosidase inhibitors [4]. α-Amylase (EC 3.2.1.1) is an enzyme that plays a crucial role in the digestion of carbohydrates. It is produced in various organs, including the salivary glands, pancreas, and small intestine. The main function of α-amylase is to break down complex carbohydrates, such as starch and glycogen, into smaller, more easily digestible molecules. It catalyses the hydrolysis of the α-1,4 glycosidic bonds that hold the glucose units together in these polysaccharides. This process results in the production of shorter chains of glucose called dextrin. α-Glucosidase (EC 3.2.1.20) also plays a vital role in the digestion of complex carbohydrates, particularly disaccharides and oligosaccharides. It is produced in the small intestine and is responsible for breaking down these complex carbohydrates into simple sugars for absorption.

The peculiar properties of common bean inhibitors have made possible the production of dietary supplements commonly called “starch blockers”. The administration of “starch blockers”, marketed under different names such as PHASE2^®^, to overweight and diabetic subjects contributes to weight loss and glycaemic index control [4]. The effectiveness and safety of these kinds of dietary supplements and the need for alternative and natural treatments for these health conditions have resulted in an impetus to search for plant-derived drugs [5].

However, the raw bean extract also contains a variety of antinutritional factors, such as trypsin inhibitors, phytic acid, saponins, tannins, and lectins such as phytohemagglutinin (PHA), that could be potentially harmful to human health [6]. PHA acts by agglutinating red blood cells in mammals, so its presence is very undesirable. Although steam cooking reduces PHA activity [7], the thermal treatment also reduces amylase activity [8].

Therefore, it is very important to identify cultivars with high α-AI activity and low concentrations of PHA as starting materials to produce food supplements.

In the present research, ten Italian *P. vulgaris* cultivars were screened for bioactive (α-amylase and α-glucosidase inhibitors) and antinutritional (PHA) compounds.

Nine cultivars came from different Sardinian villages throughout the region. Sardinia is an island located in the central–western part of the Mediterranean Sea. It is the second-largest island in the Mediterranean, after Sicily, representing one hotspot of plant biodiversity [9]. These nine cultivars come from small plots of land, are rare, and have not yet been characterized for their α-amylase inhibiting power. The choice to investigate the Sardinian cultivar comes from the desire to preserve Sardinian biodiversity. In fact, ongoing climate change is leading to increased stress for common beans due to many biotic and abiotic factors. Preserving such nearly forgotten cultivars could also have significant economic and ecological impacts on the island, mainly due to the increased interest in consumer demand for traditional and healthy foods [10].

As a comparison, we selected another cultivar, a commercial product called Lamon, from Veneto, a region located in northeastern Italy. This bean is well known for its beneficial properties, which are responsible for its considerably high market price. It is also employed in some food supplements used for weight control.

In addition, we also focused on the structure–activity relationship of the interaction between α-AI from these cultivars and α-amylase. In fact, α-AI is a quite complex protein; several α-AI isoforms are present in the different cultivars of the legume *P. vulgaris*, all encoded by genes that are a part of a gene cluster [4]. The α-AI isoforms in bean are synthesized in the endoplasmic reticulum (ER) as inactive preproproteins to become active proteins after posttranslational modifications via *N*-glycosylation and proteolytic cleavage. These posttranslational modifications give rise to two subunits (α and β subunits), with molecular weights of 7800 Da (77 amino acid residues) and 14,600 Da (146 amino acid residues), respectively, that aggregate into a final α_2_β_2_ tetramer. The α-AI isoforms of *P. vulgaris* differ in their primary structures and *N*-glycosylation sites. Several amino acid residues of α-AI have been identified as important for inhibitory activity towards porcine pancreatic α-amylase (PPA), such as Y60, Y209, Y213, D61, and S212 arranged in two hairpin loops called L1 and L2 (the numbers refer to the positions of the residues in the preprotein).

Knowledge of the sequences and structural features of α-AI proteins is vital for a better understanding of the details of the inhibition process and its control.

Using all the sequence structure information available in gene databases, our research objective was also to construct partial structural models of α-AI in 10 *P. vulgaris* cultivars. Such models could provide insights into these inhibitors and their interactions with α-amylase enzymes.

Among the possible characterization techniques of protein–protein interactions, computer simulations based on mechanistic (i.e., structural) representations have gained popularity in recent years. This has been made possible thanks to the advent of novel computational techniques that benefit from the increasing computing power of modern supercomputers [11]. In this study, we also applied a quantum mechanical characterization technique, enabled by high-performance calculations in supercomputers, to characterize the interaction patterns of biological subunits by first-principle calculation, which enabled us to identify the hot-spot regions of the system, classify them in terms of their relevance, and characterize the interaction signature (binding phenotype) of the protein-protein systems under analysis [12]. Such a characterization can be performed completely in silico, starting from a structural representation of the system.

## 2. Results and Discussion

### 2.1. α-AI Inhibitor and α-Glucosidase Inhibitor Activity

Based on the growing need to find new plant-based bioactive compounds useful for the treatment of obesity and diabetes, ten *P. vulgaris* cultivars were tested for their inhibitory activities against digestive enzymes and for the presence of anti-nutritional factors. Many enzymes are involved in carbohydrate digestion. Before being absorbed in the final part of the intestinal tract, starch is hydrolysed by a combination of salivary and pancreatic α-amylase and α-glucosidase [13]. Considering the role of α-amylase and α-glucosidase enzymes in sugar absorption, the presence of α-amylase and α-glucosidase inhibitory activities is a desirable condition in a vegetable extract.

The comparison of raw extracts of the examined cultivars indicated that eight out of ten inhibited α-amylase porcine pancreatic enzyme, as shown in Table 1. Three varieties, namely, Fazadu Nieddu, Nieddone, and Bianco, had the highest inhibitory activity (351 ± 19, 356 ± 26, 376 ± 12 IAU/g, respectively), but no significant differences were determined among the other eight cultivars, including Lamon. This cultivar was chosen as a commercial control. In fact, its beneficial properties are well known. In addition, Lamon is currently a reference standard in Italy to produce food supplements for carbohydrate metabolism management (https://www.indena.com/product/beanblock/, accessed on 10 May 2023). Both of these features are responsible for its considerably high market price.

The comparison among the inhibitory activities highlighted the similarity of the Sardinian cultivars with Lamon in terms of inhibitory activity, so they could be competitive as a source of raw material for commercial applications.

Using human salivary amylase, some significant differences were determined, since three Sardinian cultivars, Fasolu, Faitta Sorgonese, and Fazadu Nieddu, showed even higher inhibitory power than Lamon (196 ± 5, 185 ± 30, 170 ± 3 IAU/g, respectively, Table 1). Two cultivars, Fasolu Pintau and Faitta a Cavanedda, had no α-AI inhibitor activity (both using porcine pancreatic and human saliva α-amylase). In our opinion, investigation of these cultivars could reveal crucial information about the structure–activity relationship of this inhibitor. Data from other Southern Italy common bean cultivars suggest that a similar pattern could also be present between different harvest years of the same cultivar [14]. Such pieces of information could also encourage multiple harvest-year screening of these samples.

Rather than α-AI inhibitory activity, α-glucosidase inhibitory activity was instead recorded in all cultivars. Fasolu had the highest inhibitory activity (182 ± 21 IGU/g), with significant differences reported, in comparison with the lowest active cultivars.

No significant correlation was observed between the inhibitory activity of porcine α-amylase and yeast α-glucosidase (*p* > 0.05; *r* = 0.31). This phenomenon was confirmed by the presence of α-glucosidase inhibitory activity in the two cultivars without α-AI. As previously suggested, different classes of compounds could be involved in α-AI/α-glucosidase inhibition [14,15].

Despite what was previously observed [15], in our work, no evident correlation with the seed colour and the inhibitory activity was observed; in fact, the light-coloured cultivar Bianco, for instance, showed low inhibitory activity against human α-amylase but high activity against porcine pancreatic α-amylase. In contrast, Fasolu, which had the lowest inhibitory activity against porcine α-amylase, showed the highest activity against human α-amylase. The other cultivar with high inhibitory activity against porcine α-amylase (Nieddone) is black in colour.

On the other hand, the other cultivars with great inhibitory activity against human α-amylase (Faitta Sorgonese and Fazadu Nieddu) are white and spotted white/black in colour.

More investigations are needed to verify whether the colour of the outer seed tegument influences the inhibitory activity, but an earlier study seems to confirm the absence of a correlation [16].

The total protein content ranged from 21.9 ± 2.6 mg/g in Fasolu Pintau to 35.4 ± 1.7 mg/g in Bianco (Table 1). This variability could be related to genetic diversity or different agro-techniques and different *P. vulgaris* growing conditions. No correlation was found between the total protein content and the inhibition of α-amylases and α-glucosidase (*p* > 0.05, 0.04 < *r* < 0.16), in accordance with what was observed with other bean cultivars [15]. However, it is interesting that despite the zero values of Faitta a Cavanedda and Fasolu Pintau, a correlation between total protein and porcine α-amylase inhibition occurred (*p* < 0.05, *r* = 0.79). The other *p* values remained greatly above 0.05.

The presence of α-amylase inhibition activity has been reported in different common bean varieties—pinto [17], white bean [18], and red kidney bean [19]—but to date, no studies have been conducted on the inhibitory power of extracts of these Sardinian varieties. Other Mediterranean and southern Italian common bean cultivars have been studied for their action on digestive enzymes and for the role of polyphenol content in the inhibition of α-amylase and α-glucosidase [15,20].

A quantitative comparison of the findings of the present study with those of these previous studies is not always possible since different methods and protocols have been employed in the studies reported in the literature. We preferred to avoid the most commonly used protocol (involving reducing sugar measurement with 3,5-dinitrosalicylic acid reagent) because many commercial enzymes are contaminated by lactose used as a preservative.

The inhibitory power on α-amylase and α-glucosidase could also be due to the presence of polyphenols, as already shown in Mexican and Brazilian cultivars, cranberry beans, and black turtle beans [21,22,23]. However, a boiling treatment of our extracts for 10 min eliminated the inhibitory power, confirming that this is mostly likely due to proteins.

In this respect, however, the study of polyphenol content in Sardinian beans is useful for a better characterization of Sardinian cultivars (even if the focus of the present study was limited to activity from protein inhibitors).

### 2.2. Haemagglutination Activity

Haemagglutinins belong to the lectin family, a heterogeneous group of glycoproteins resistant to intestinal proteolysis that agglutinate blood cells [24]. These proteins are contained in raw beans, but their activity can be eliminated by heat [25]. Since common beans are usually consumed after cooking, no side effects occur after their ingestion. However, it is necessary to pay particular attention to the residual lectin activity present in food supplements [26]. In this respect, there is a clear interest in identifying cultivars with low haemagglutination power.

With this aim, human red blood cell agglutination was performed with serial dilutions of our extracts, identifying the minimum dilution able to still agglutinate cells (MAC).

Previous studies have shown that lectins from the family Fabaceae usually cause agglutination. Three plant species, *Pisum sativum*, *Phaseolus vulgaris*, and *Glycine max*, caused agglutination of erythrocytes from all blood groups [27].

All screened cultivars tested in this study had haemagglutination activity, even with significant dilution (average value 12.5 mg/mL). Only Nieddone showed no activity, even at the highest concentration tested, which was 200 mg/mL (Table 2).

This desirable condition has also been reported for other *P. vulgaris* cultivars: Great Northern [15], Tapiramo, and Mottled bean [28]. Other seeds from members of the Fabaceae family, such as *Arachis hypogea*, have no haemagglutination activity [29]. Particular attention must be paid to the case of *Glycine max*. Notably, haemagglutination activity was recorded in soybean by some authors [28,29], but not by others [15]. This discrepancy could be related to differences in the soybean cultivars investigated.

Typically, only white common bean extract is used for commercial supplements [4], and a previous study found that the coloured common beans had the highest haemagglutination power [30]. However, the data here-reported show that black-coloured beans may also not have these antinutritional factors.

The absence of haemagglutination power and the presence of α-AI/α-glucosidase inhibitors makes Nieddone the most interesting cultivar as starting material to produce lectin-free food supplements or commercial lectin-free product in accordance with what has been suggested by other studies [31].

### 2.3. PCR Analysis of cDNA Obtained from P. vulgaris Cultivars

The molecular results, as well as the expansion of the genetic information regarding this plant, could add a new piece to the puzzle of the knowledge concerning the primary structure of the α-AI protein, through its isolation, and of the transcript coding for the α- and β-subunit of the protein in all ten cultivars.

By exploiting the high amino acid sequence homology among α-AI proteins of the genus *Phaseolus*, we designed partially degenerate primer pairs using the CODEHOP bioinformatics program. Using the pairs of primers available on the synthesized cDNAs, we obtained sequenceable fragments that allowed us to reconstruct part of the nucleotide sequence of the α-AI gene for each cultivar. The *P. vulgaris* cultivar α-AI gene sequences obtained in this study were deposited in NCBI GenBank, as shown in Table 3.

We reconstructed approximately 62% of the entire nucleotide sequence of pv28-α-AI from the Granino cultivar and approximately 30% of those of the other 9 cultivars. The nucleotide sequences obtained, compared with those deposited in the NCBI GenBank and in the EMBL-EBI databases (https://www.ebi.ac.uk/, accessed on 5 April 2023), greatly resembled several plant α-AI sequences. The greatest degree of similarity emerged from the alignment with *P. vulgaris* α-AI-1 (Accession No. CAD28835) and *P. vulgaris* α-AI-4 (Accession No U84390.1) isoforms. Once in silico translation of the nucleotide sequence into amino acids had been achieved, the primary structure obtained for each cultivar was aligned with the proteins deposited in the UniProt database (https://www.uniprot.org/uniprotkb/, accessed on 5 April 2023) to obtain information about the homology with the β-subunit of α-AI-1 (Accession No P02873.1) and α-AI-4 (Accession No AAB42070.1) of *P. vulgaris*.

As shown in Figure 1, PV28-α-AI was the protein from which we obtained most of the information on the primary structure based on 152 amino acids. Its sequence aligned from amino acid 91 to amino acid 243 of P02873 α-AI-1 and from amino acid 89 to amino acid 241 of the AAB42070.1 α-AI-4 isoform (the numbering refers to the preprotein).

Regarding the other nine proteins, we obtained 77 amino acids from the whole α-AI sequence. Eight of the ten sequences (i.e., all except for PV20-α-AI and PV152-α-AI), aligned from amino acid 129 to amino acid 205 of the P02873 α-AI-1 protein and from amino acid 127 to amino acid 203 of the AAB42070.1 α-AI-4 isoform. Specifically, for the proteins PV177-α-AI, PV147-α-AI, PV124-α-AI, PV121-α-AI, PV113-α-AI, PV1-α-AI, and PV-LAM-α-AI, we obtained information about the primary structures that showed very high sequence identity compared to β subunits of α-AI-1 (94%) and α-AI-4 (89%) of *P. vulgaris* isoforms (Figure 2A,B), whereas for PV152-α-AI and PV20-α-AI, the highest homology value was less than 50%.

To better highlight variations in the primary structures of PV152-α-AI and PV20-α-AI, we aligned their amino acid sequences with the primary structures of the α-AI-1 and α-AI-4 isoforms. The alignment with the α-AI-1 isoform showed the presence of two gaps of 11 amino acids, extending from amino acids 161–172 and 189–200, and an identity of approximately 33% (Figure 3A). Moreover, we observed that PV152-α-AI and PV20-α-AI primary structures were more similar to isoform α-AI-4. In fact, aligning their amino acids revealed a homology percentage of approximately 40% with the isoform α-AI-4 and only one gap in the sequence (Figure 3B).

### 2.4. Amino Acid Pattern and Profile Search Results

The α-AIs of *P. vulgaris* generally differ slightly in their primary structures and in their *N*-glycosylation sites. α-AI-1 was the first isoform of *P. vulgaris* to be isolated and molecularly characterized [32,33]. In the endoplasmic reticulum, the preprotein undergoes cleavage of the signal peptide and *N*-glycosylation of residues of Asn35, Asn88, and Asn163 (the numbers refer to the positions of the residues in the preprotein). The prediction of glycosylation sites was obtained for other *P. vulgaris* inhibitor isoforms, confirming previous results obtained for α-AI-1 [34]. In this study, a search for potential glycosylation features was conducted using the NetNGlyc tool. Considering the partial sequences obtained for all cultivars, our results are consistent with the finding that at least one potential *N*-glycosylation site occurred at the same amino acid position of α-AI-1 in eight of the deduced cultivar sequences. In contrast, no potential sites were found in the PV152-α-AI and PV20-α-AI sequences (Figure 4). In addition to the potential *N*-glycosylation sites, protein kinase C phosphorylation sites were found in eight of the deduced sequences using the NetPhos-3.1 tool. In PV152-α-AI and PV20-α-AI, in which no potential sites for protein kinase C were found, potential phosphorylation sites (PRRSS) for protein kinase A (PKA) were present instead (Figure 4).

We also investigated the amino acid residues important for the inhibitor activity but found evidence only for the PV28-α-AI protein, for which we determined almost the complete sequence of the β subunit. The amino acid residues Y209, S212, and Y213 of the β subunit (the numbering of the amino acids refers to the preprotein), important for the interaction with the α-amylase enzyme, are conserved in the PV28-α-AI protein (Figure 4).

### 2.5. Deduced Phylogenetic Analysis and Secondary Structure

α-AI-1 (P02873.1) and α-AI-4 (AAB42070.1) sequences from *P. vulgaris* were used as out-groups. Phylogenetic analysis confirmed a notable similarity between the predicted amino acid sequences of PV152-α-AI and PV20-α-AI and the isoform α-AI-4 with respect to isoform α-AI-1, which was closely related to the other eight cultivars (Figure 5).

The in silico reconstruction of the secondary structures showed differences in the distribution of the α-helices, β-sheets, and coils of the PV152-α-AI and PV20-α-AI proteins compared with the proteins deduced from the other cultivars (Figure 6).

However, several factors could explain the absence of inhibition towards α-amylase observed in the ACC 113 and ACC 147 cultivars. Since the complete primary structure of these proteins was not available, it proved impossible to establish whether the amino acid residues important for protein maturation and activity are conserved [4]. Another explanation could be the presence of microRNAs (miRNAs) in their cells that were able to inhibit translation of α-AI. Usually, plant miRNAs associate with Argonaute protein 1 to promote RNA posttranscriptional gene silencing by coupling target sequences and determining RNA slicing and/or translation inhibition [35,36]. In the last few years, many miRNAs have been identified in *P. vulgaris* [37,38]. Based on this second hypothesis, the α-glucosidase inhibitory activity observed in the cultivars without α-AI activity could be linked to the presence of flavonol glycosides and caffeic acid derivatives in the homogenates. Recent research has demonstrated that various *P. vulgaris* (Chilean bean) extracts were strongly active against α-glucosidase but were inactive towards α-amylase [39]. These hypotheses will be investigated to gain further knowledge of the primary structure of α-AIs and to investigate the possible presence of antisense miRNAs and phenolic substances that are active against α-glucosidase.

### 2.6. Mechanistic Characterization of the Interaction between α-Amylase and α-AI

To understand whether these structural differences highlighted by molecular analysis could influence the protein–protein interaction between α-amylase and α-AI, we also performed an in silico analysis. The contribution of each residue to the overall binding performance was calculated, highlighting which amino acids facilitate or hinder binding and how, using the 1DHK PDB porcine pancreatic α-amylase enzyme and α-AI (P02873) amino acid sequence as a model. In fact, this sequence has already been used as a reference in several literature studies [40,41,42].

We investigated the electronic density around the protein using the QM-CR DFT method described below to identify the interface residues and chemical hotspot interactions.

Once the chemical connection among amino acids was identified, we assigned to each residue its contribution to the binding interaction between the two subsystems. We calculated these interaction terms from the output of the DFT code and interpreted them as two parts: first, a long-range electrostatic attraction/repulsion term, defined from the electron distributions of each of the fragments (even when far apart, two fragments may still interact); the remaining term, which can only be attractive, is provided by the chemical binding between the fragments and is nonzero only if the electronic clouds of the fragments superimpose (short-range).

By including long-range electrostatic terms, the decomposition enabled us to single out relevant residues not necessarily residing at the interface. In this way, the model provides an ab initio representation of the RBD–ligand interactions as the final output.

Mechanistic characterization of the binding between α-amylase and its inhibitor from the bean *P. vulgaris* is shown in Figure 7. The data are plotted on the sequence of the bean. Letters represent single amino acid residues. “Å” is the distance of a residue to the nearest atom of its ligand. “Tot” is the chemical/electrostatic force shown as attractive (blue) or repulsive (red), with darker colours indicating stronger effects.

The “Tot” interaction is the sum of “vdW”, “H”, and “El” quantities, which correspond to the van der Waals forces and chemical (steric) and electrostatic interactions, respectively, and they are shown in the separated lines above the “Tot” line.

We identified two main hotspots corresponding to aspartic acid D38 and tyrosine Y186 (the numbering of the amino acids refers to the mature protein). The interaction is mainly governed by the chemical contribution from the closer amino acids. The values of the interaction are shown in the bottom panel of Figure 7.

Figure 8 shows the emerging interaction network that connects the relevant residues of the α-AI sequence from *P. vulgaris* (P02873) with the corresponding interface residues of the porcine pancreatic amylase. The network is drawn by following the FBO between amino acids of the respective sequence. Thicker lines indicate stronger chemical bonds. Nodes are coloured following the total interaction values, and their border is based on the fragment bond order (FBO) values at the interface, as per Figure 8.

Figure 9 highlights the relevant part of the general interaction graph of Figure 8.

Unfortunately, the two hotspots (D38 and Y186) were not yet identified and sequenced by the CODEHOP strategy in our cultivars. Only in the case of cultivar Granino (PV28-α-AI) was an amino acid fragment corresponding to almost all β subunits reconstructed, including the Y186 position. However, this reconstructed sequence of the Granino cultivar showed no differences from the amino acid reference sequence used for our analyses (P02873.1).

On the contrary, for the cultivars other than Granino, it was not yet possible to gain information about these hotspots. Furthermore, it was not possible to obtain information on the contribution of the amino acids of PV152-α-AI and PV20-α-AI positioned upstream and downstream of the gaps resulting from the alignment to the homologous sequences of α-AI-1 and α-AI-4 (Figure 3A,B). In particular, the amino acids of the two Sardinian cultivars placed downstream of the second gap (Figure 3A) and of the single gap (Figure 3B) showed conserved substitutions in the alignment with the amino acids Y209-S212-Y213 of α-AI-1 and S210-Y212 of α-AI-4, located in the hairpin loop L2 (the numbering of the amino acids refers to the preprotein). Despite the differences in the primary structure observed compared to the reference isoforms α-AI-1 and α-AI-4, the two cultivars showed inhibitory activity towards α-amylase. This observation, which should be investigated in future studies, could be important for opening new hypotheses on the functional role of the amino acids of the β-subunit.

However, this can be an important incentive to complete the sequences from our samples and thus obtain complete information about sequence–structure–activity relationships. Such knowledge can be crucial to effectively screen unknown varieties or to develop new recombinant α-AI sequences with improved and modulated features.

## 3. Materials and Methods

### 3.1. Chemicals

2-Chloro-4-nitrophenyl-α-d-maltotrioside (93834-100MG), p-nitrophenyl-α-d-glucopyranoside (487506-1GM), α-amylase from porcine pancreas (Type VI-B A3176-2.5MU), and α-glucosidase from *Saccharomyces cerevisiae* (Type I G5003-1KU) were purchased from Sigma-Aldrich (Milan, Italy). All other reagents used were of the highest grade available, purchased from Sigma-Aldrich (Milan, Italy), and used without further purification.

### 3.2. Plant Materials

The cultivars included in this study are part of a wider germplasm collection of the Sardinian Agricultural Agency (AGRIS, Cagliari, Italy), the Regional Agency for the implementation of agricultural and rural development programs (LAORE), and the Centre for the Conservation and Enhancement of Plant Biodiversity (CBV; Sassari; Italy). Nine common beans (*Phaseolus vulgaris* L.) were collected in all regions of Sardinia, mainly from local small farms, and differentiated by morphologic features (different shape, dimension, and colour), as shown in Table 4 and Figure 10. 

One Venetian commercial cultivar (Lamon) was also included in the work as a comparison model. Lamon was purchased from the local Consortium for the Protection of the Protected Geographical Indication (http://www.fagiolodilamon.it/en/, accessed on 5 April 2023).

As an experimental positive control, CarboStop commercial food supplement (LongLife Nutritional Supplements, Peschiera Borromeo, Milan, Italy) was purchased from a local market. The supplement is based on a purified common bean extract. Each tablet contains 500 mg of extract and was processed using a similar protocol to that used for *P. vulgaris* seeds.

### 3.3. Protein Extraction

Legume seeds were ground with a grinder, and the resulting powder was suspended in Bis–Tris buffer (pH 6.5 0.1 mM) and 0.1 M NaCl and was stirred for 1 h and centrifuged at 12,000 rpm for 10 min at 4 °C. The supernatant was analysed to record biological activities. Total proteins were measured by the Bradford method using bovine serum albumin as a standard [43].

### 3.4. α-Amylase Inhibition Assays

The inhibition of α-amylase activity was also determined using 2-chloro-*p*-nitrophenyl-α-d-maltotrioside (CNP-G3), as previously described with minor modifications [44]. A reaction mix containing 250 mM sodium phosphate buffer at pH 6.5 with 60 mM NaCl, 5 mM CaCl_2_, 500 mM KSCN, and 5 E.U. α-amylase from porcine pancreas (unless otherwise stated) was used. The solution was mixed and incubated in the absence or presence of samples at 37 °C for 10 min in a final volume of 150 μL. After incubation, 50 μL of a 9 mM CNP-G3 solution was added, and the amount of 2-chloro-*p*-nitrophenol released by enzymatic hydrolysis was monitored at 405 nm in a UV–VIS MultiskanGo microplate spectrophotometer (Thermo Fisher Scientific, Monza, Italy). Proper negative controls were used in the absence of enzyme, sample, or substrate. As a positive control, a commercial food supplement extract (based on *P. vulgaris*) was added for each analysis.

One amylase E.U. was defined as the amount of enzyme capable of hydrolysing 1 μmol of CNP-G3 per minute at pH 6.5 and 37 °C (monitoring CNP formation, ε_405_ = 14,580 M^−1^ cm^−1^). The amylase inhibitory unit (IAU) was defined as the number of amylase units inhibited under the assay conditions.

### 3.5. α-Glucosidase Inhibition Assays

The α-glucosidase inhibition assay was conducted under the same experimental conditions as the CNP-G3 protocol. The substrate used was 2.25 mM *p*-nitrophenyl-α-d-glucopyranoside (pNPG). One glucosidase E.U. was defined as the amount of enzyme capable of hydrolysing 1 μmol of pNPG per minute at pH 6.5 and 37 °C (monitoring CNP formation, ε_405_ = 14,580 M^−1^ cm^−1^). The glucosidase inhibitory unit (IGU) was defined as the number of glucosidase units inhibited under the assay conditions. Proper negative controls were performed in the absence of enzyme, sample, or substrate. As a positive control, a commercial food supplement extract (derived from *P. vulgaris*) was added in each analysis.

### 3.6. Haemagglutination Assay

Human blood (2 mL, collected from healthy volunteers) was centrifuged at 2000× *g* for 3.5 min, the supernatant was discarded, and the cell pellet was washed three times with PBS buffer (137 mM NaCl, 2.7 mM KCl, 8 mM Na_2_HPO_4_, 1.5 mM KH_2_PO_4_, pH 7.4). The red blood cell (RBC) pellet was suspended to 50% (*v*/*v*) with PBS buffer, stabilized with 0.01% sodium azide and stored at 4 °C. The lifespan of the RBCs was up to three weeks. One aliquot of 10% RBC was prepared, and each raw extract was used at different serial dilutions (pure, ½, ¼, 1/8, 1/16, …, down to 1/512.)

For each combination of 10% RBC and protein concentration, the RBC sample was pipetted onto a glass microscopy slide and mixed in a 10 µL:10 µL ratio with the protein solution. The mixture was incubated for 5 min and observed by light microscopy using an Olympus BX81 digital microscope (Olympus, Segrate, Italy). Images were captured with a charge-coupled device camera (Cohu, San Diego, CA, USA). Control samples were used by mixing 10 µL of the erythrocyte suspension with 10 µL of the PBS buffer, which was incubated with RBCs. The degree of agglutination was recorded in relation to the status of an absence of agglutination, negative reactions appeared as unchanged cloudy suspensions, and several large aggregates and visible aggregations of particles were seen in the case of positive reactions. In the case of negative agglutination after 5 min, the incubation was extended up to one hour. The results are expressed as the minimum protein concentration able to agglutinate the sample (MAC, mg/mL).

### 3.7. Biomolecular Assay: Plant Materials

*Phaseolus vulgaris* leaves obtained from seeds of the cultivars described previously were the starting material for RNA biomolecular experiments. The tissue was treated with liquid nitrogen to obtain a delicate powder and was subsequently frozen and stored at −80 °C until use.

### 3.8. RNA Extraction and Reverse Transcription

Total RNA from *P. vulgaris* for RT-PCR RNA was extracted from leaves previously treated with liquid nitrogen using TRI Reagent (Sigma-Aldrich, St. Louis, MO, USA) following the manufacturer’s recommended protocol [45]. The quality of the purified RNA was verified using gel electrophoresis with a 1% denaturing agarose gel stained with SYBR Green II (Sigma-Aldrich, St. Louis, MO, USA), and the concentrations were measured using a NanoDrop 2000c UV–VIS spectrophotometer (Thermo Scientific, Waltham, MA, USA) at 260 nm. To obtain cDNAs, RNAs were reverse transcribed with an oligo dT primer using an enhanced avian myeloblastosis virus reverse transcriptase enzyme (Sigma-Aldrich, St. Louis, MO, USA), following the manufacturer’s recommendation.

The following components were mixed in a sterile microfuge tube: 2 μL 10× reaction buffer; 4 μL MgCl2, 25 mM; 2 μL deoxynucleotide mix 40 mM; 2 μL oligo-p(dT)15 primer (0.8 μg/μL); 1 μL RNase inhibitor 50 U/μL; 1 μg of total RNA; and PCR-grade water to a final volume of 20 μL. All components were supplied by Sigma-Aldrich. The reaction was incubated at +25 °C for 10 min and then at +42 °C for 60 min. Following the +42 °C incubation, the AMV reverse transcriptase was denatured by incubating the reaction at +99 °C for 5 min, and then the reaction was cooled to +4 °C for 5 min.

### 3.9. PCR-CODEHOP Strategy

To detect the unknown nucleotide sequences of the α-AI from the different *P. vulgaris* cultivars, a degenerate hybrid oligonucleotide primer (CODEHOP) strategy was used [46], which began with aligning the multiple sequences of α-AI proteins using the Clustal Omega program (http://www.ebi.ac.uk/clustalo, accessed on 6 March 2023) to show the degree of homology among the species considered.

Three different plant sources were chosen from the GenBank SWISSPROT database (*P. vulgaris* accession no. CAD28835, *Phaseolus costaricensis* accession no. CAH60260.1, and *Phaseolus coccineus* accession no. CAH60259.1). After alignment (Figure 11), the sequences were pasted into a block of the motif finders using EMBL-EBI search and sequence analysis tools [47]. The primers were designed using the default parameters of the j-CODEHOP server (https://4virology.net/virology-ca-tools/j-codehop/, accessed on 25 March 2023). The primers for amplifying *P. vulgaris* cultivar α-AI cDNAs were chosen from a group of primer candidates provided by the j-CODEHOP program (Table 4). Briefly, the first fragment of partial α-AI cDNA was obtained using the sense primer F2 (5′-GCTGGTGCCCGTGCAGCCCAAGTCCaarggngayac-3′) together with the antisense primer R2 (5′-TCAGCACGATGTTGGACCGCTCGGAyttytgrtcytt-3′) (Table 5).

Each primer representing the consensus clamp is uppercase, whereas the degenerate core is lowercase: y = [C,T]; r = [A,G], and n = [A,G,C,T]. PCR was performed in a solution containing 1.5 mM MgCl2, 100 mM Tris-HCl, pH 8.3, 50 mM KCl, 200 mM dNTP mix, 1 mM sense primer, 1 mM antisense primer, 100 ng cDNA, and 1–3 units of Jump Start AccuTaq LA DNA polymerase mix (Sigma-Aldrich, St. Louis, MO, USA). The thermal amplification cycles were conducted in a Personal Eppendorf Mastercycler (Eppendorf, Hamburg, Germany) using slightly different programmes. The CODEHOP cDNA PCR products with the expected sizes were purified with a Charge Switch PCR Clean-Up kit (Invitrogen, Carlsbad, CA, USA) and then sequenced. The PCR products were obtained under the following thermal cycling steps: 30 cycles of 94 °C for 30″, 45 °C for 30″, and 72 °C for 30″.

### 3.10. cDNA Sequencing and Analysis

cDNA sequencing was performed by Bio-Fab Research (Rome, Italy). Nucleotide and deduced amino acid sequence analyses were performed using freeware programs. The translation of nucleotide sequences was performed using ExPASy Translate routine software (http://ca.expasy.org/, accessed on 4 March 2023). Similarities were analysed with the advanced BLAST algorithm, available at the National Centre for Biotechnology Information website (http://www.ncbi.nlm.nih.gov/, accessed on 7 April 2023), and with the FASTA algorithm version 3.0 from the European Bioinformatics Institute website (http://www.ebi.ac.uk/fasta33/index, accessed on 8 April 2023). Sequences were aligned using Clustal Omega software 1.2.2.

### 3.11. Amino Acid Pattern and Profile Search

The NetNGlyc tool (https://services.healthtech.dtu.dk/service.php?NetNGlyc-1.0, accessed on 14 May 2023) was used to detect potential *N*-glycosylation sites. The tool assumes that *N*-glycosylation occurs on asparagine, which is localized in the Asn–Xaa–Ser/Thr stretch (where Xaa is any amino acid except proline). Although this consensus tripeptide (also called an *N*-glycosylation sequon) is needed, it is not always sufficient for asparagine to be glycosylated [48]. Any potential crossing of the default threshold of 0.5 represents a predicted glycosylated site. The potential score obtained is the averaged output of nine neural networks.

The NetPhas-3.1 tool (https://services.healthtech.dtu.dk/service.php?NetPhos-3.1, accessed on 14 May 2023) was used to detect potential kinase phosphorylation sites [49].

### 3.12. Phylogenetic Tree Construction

The α-AI partial cDNA sequences were submitted for phylogenetic analysis. The phylogenetic tree was constructed using Mega 11 software [50]. Evolutionary history was inferred using the neighbour-joining method [51]. The bootstrap consensus tree inferred from 500 replicates was taken to represent the evolutionary history of the taxa analysed [52]. Branches corresponding to partitions reproduced in less than 50% bootstrap replicates were collapsed. The percentage of replicate trees in which the associated taxa clustered together in the bootstrap test (500 replicates) was shown next to the branches. The consistency of the inferred phylogenetic tree was evaluated using bootstrap analysis with 500 replications.

### 3.13. Secondary Structure Prediction

The partial amino acid sequence coding for the α-AI of each *Phaseolus vulgaris* cultivar was subjected to secondary protein structure prediction using the Chou–Fasman method [53]. The tool used is available at http://cib.cf.ocha.ac.jp/bitool/MIX/, accessed on 21 March 2023.

### 3.14. Complexity-Reduction Quantum Mechanical Calculation for α-Amylase and α-AI Interaction

A recently developed approach for large-scale electronic structure calculations was applied: complexity reduction in density functional theory (DFT) calculations [12,54], hereafter called quantum mechanical complexity reduction (QM-CR). This approach allowed us to study and define protein–protein interactions, thus highlighting α-AI hotspots necessary for α-AI inhibition activity.

QM-CR differs from previous approaches in requiring no targeted parameterization or prior knowledge about the nature or sites of interactions, and it is based on full QM calculations on the entire system.

QM-CR leverages recent progress in computational chemistry [11] to handle tens of thousands of atoms in a single simulation. This enables us to capture and investigate biological processes involving several hundred amino acids.

Importantly, QM-CR can reveal the mechanisms behind intermolecular binding by decomposing interactions into chemical/short-range interactions (which imply a shared electron) versus electrostatic/long-range interactions (which do not involve shared electrons). We defined “hotspots” as amino acids with significant chemical contributions to the intermolecular interactions.

The BigDFT computer program [55] was employed based on an ab initio DFT approach on a set of fully atomistic 3D structural models to simulate intermolecular interactions of interest with a computational cost manageable on modern supercomputers.

The approach employs the formalism of Daubechies wavelets to express the electronic structure of the assemblies in the framework of the Kohn–Sham (KS) formalism of DFT [55]. The electronic structure is expressed by both the density matrix and the Hamiltonian operator in an underlying basis set of support functions—a set of localized functions adapted to the chemical environment of the system.

The code provides efficient and accurate QM results for full systems of large sizes, delivering excellent performance on massively parallel supercomputers.

In the present study, the PBE approximation corrected by dispersion D3 correction terms [56] and Hartwigsen–Goedecker–Hutter (HGH) pseudopotentials [57] were employed.

Crystallographic structures were obtained from the RCSB database (RCSB PDB, accessed on 23 October 2021, https://www.rcsb.org/) using PDB entries 1bvn, 1dhk, and 1b2y.

Starting from a representative 3D model of the molecules as our input, we calculated the system’s electronic structure, from which we extracted various quantities. We drew a contact network to identify relevant chemical interactions among the spike RBD and the various interactors considered in this study. The strength of the inter-residue interaction was quantified by the “fragment bond order” (FBO), a quasi-observable parameter computed directly from the density matrix. A more detailed description of this quantity can be found in the literature [58]. The interaction is calculated using the electronic structure of the system in proximity to a given residue.

Protonation of histidine and other titratable residues was assigned a pH of 7 based on the PDBFixer tool in OpenMM (OpenMM, accessed on 23 October 2021, https://github.com/openmm/pdbfixer).

### 3.15. Statistical Analysis

GraFit 7.0.3 (Erithacus Software, London, UK), R 2.5.1 (R Foundation for Statistical Computing, Vienna, Austria), and GraphPad INSTAT 8.2.1(GraphPad Software, San Diego, CA, USA) were used for data analysis. One-way analysis of variance (ANOVA) and the Bonferroni multiple comparison test were used to assess the statistical significance of the differences. All analyses were performed at least in triplicate (unless otherwise stated), and the data are reported as the mean ± standard error of the mean (SEM).

## 4. Conclusions

The screening of nine rare Sardinian common bean varieties and of one commercial cultivar from Veneto allows us to discover and highlight some features useful to produce food supplements recommended in the treatment of obesity and diabetes.

This study showed that only two of the tested cultivars (Fasolu Pintau and Faitta a Cavanedda) are unsuitable as raw materials to produce food supplements due to the absence of inhibitory activity against porcine and, above all, against human amylases, while all cultivars have α-glucosidase inhibitory activity.

The cultivar Nieddone is the most interesting variety due to its inhibitory activity against both α-amylase and α-glucosidase enzymes in its extract, but most importantly, it lacks haemagglutination activity, an antinutritional factor that prevents the consumption of raw beans.

Molecular studies conducted on reverse transcribed RNA in cDNA showed that α-AIs were expressed in all the analysed cultivars. Having deduced fragments of the primary structure of the proteins, we speculate that the proteins PV152-α-AI and PV20-α-AI share more similarity with isoform α-AI-4 than with isoform α-AI-1, which could be reflected in a greater sharing of structural features. A future goal will be to complete cDNA sequencing to increase the understanding of the primary structure of these proteins. In fact, the mechanistic characterization suggests that the α-AI had two main hotspots in the interaction with the α-amylase enzyme, but only for the cultivar Granino (ACC.28) has an amino acid fragment that includes one of these sites been reconstructed. The PV28-α-AI does not differ from the amino acid reference sequence used for the analyses (P02873.1) in the Y186 position.

However, amino acid differences could be present at the level of the first hotspot site (D38) not yet identified in this work. These differences could explain, for example, why the cultivars Fasolu Pintau and Cavanedda have no inhibitory activity against α-amylase, although the α-AI is expressed in both.

The achievement of this objective could be important both for increasing the biomolecular knowledge of these proteins and for the development of further studies in the biotechnological field aimed at the expression of more promising peptides or proteins from a biochemical point of view.

In addition, to the best of our knowledge, this study is the first investigation of the biological activity of these nine rare Sardinian *P. vulgaris* cultivars. Such data as those from our study could represent an important landmark for the valorisation and preservation of local agronomic biodiversity.

## Figures and Tables

**Figure 1 plants-12-02918-f001:**
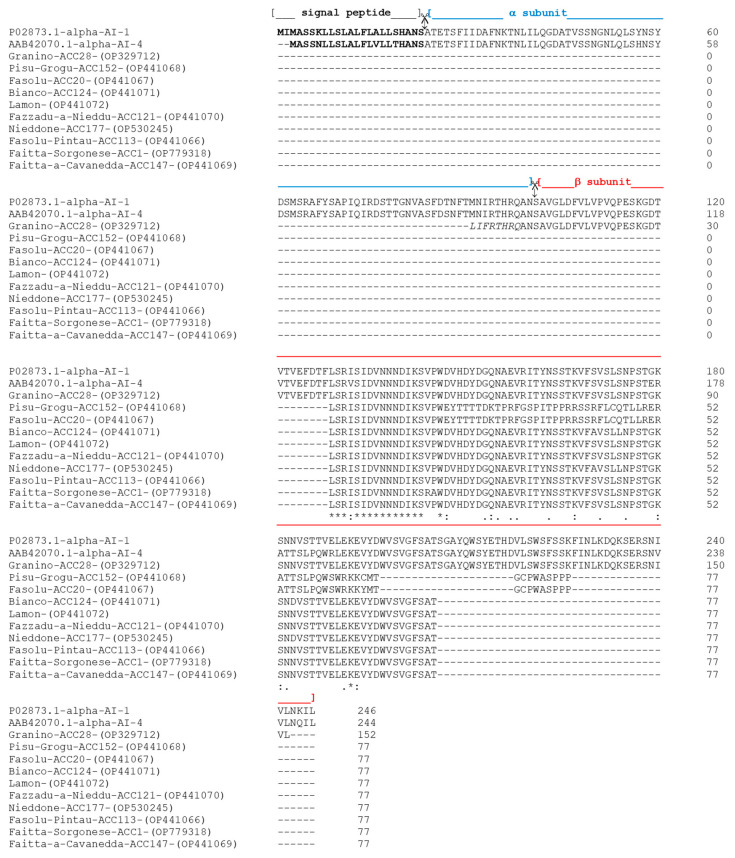
Alignment of α-AI amino acid sequences from different *Phaseolus cultivars* with homologous α-AI-1 (P02873.1) and α-AI-4 (AAB42070.1) α-AI isoforms of *P. vulgaris* from GenBank. The numbers on the left indicate the position of the amino acids in each protein. An asterisk (*) denotes identical residues; double dots (:) represent a conserved residue substitution, and a single dot (.) shows partial conservation of the residue. The figure shows the proteolytic cuts (symbol 
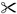
) that the preprotein underwent to give rise to the α and β subunits. Blue and red brackets indicate α and β subunits, respectively.

**Figure 2 plants-12-02918-f002:**
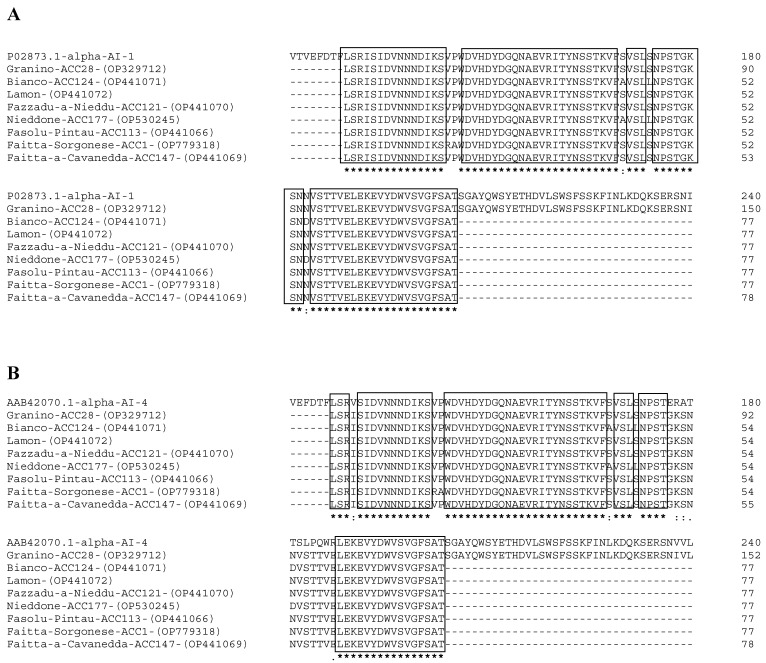
(**A**) Alignment of eight α-AI cultivar amino acid sequences from different *Phaseolus cultivars* with homologous α-AI-1 (P02873.1) and α-AI-4 (AAB42070.1) α-AI isoforms of *P. vulgaris* from GenBank. The numbers on the left indicate the position of the amino acids in each protein. An asterisk (*) denotes identical residues (94%); double dots (:) represent a conserved residue substitution; a single dot (.) shows partial conservation of the residue. (**B**) Alignment of the same sequences with the α-AI-4 (Accession No AAB42070.1) of *P. vulgaris*. The identical residues value was 89%. Alignments (**A**,**B**) show the last 77 of the 152 amino acids obtained for the PV28-α-AI sequence. Black boxes indicate conserved regions.

**Figure 3 plants-12-02918-f003:**
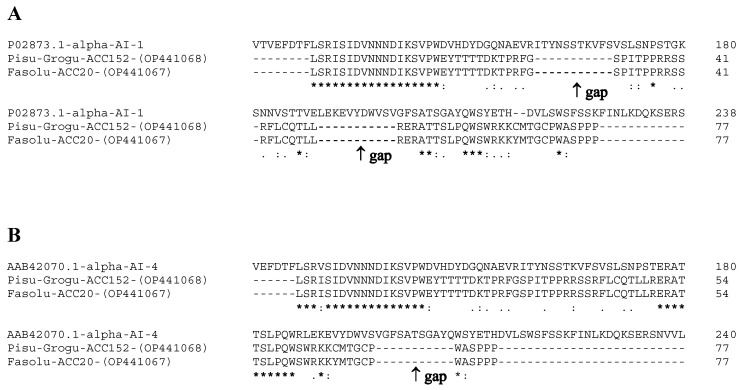
(**A**) Alignment of amino acid sequences of PV152-α-AI and PV20-α-AI with α-AI-1 isoform of *P. vulgaris*. The numbers on the left indicate the positions of the amino acids in each protein. An asterisk (*) denotes identical residues (33%); double dots (:) represent a conserved residue substitution, and a single dot (.) shows partial conservation of the residue. The gaps due to the alignment with the α-AI-1 isoform are shown with an arrow (↑). (**B**) Alignment of the same sequences with the α-AI-4 of *P. vulgaris*. The identical residues value was 40%.

**Figure 4 plants-12-02918-f004:**
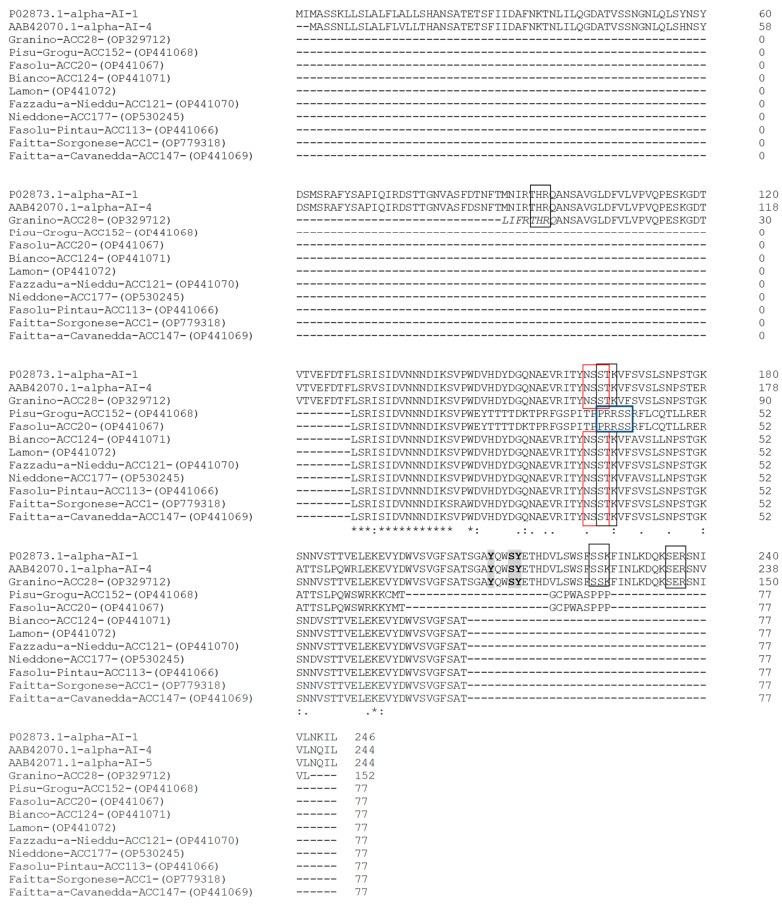
Alignment of α-AI amino acid sequences. Red boxes indicate *N*-glycosylation consensus sequences, and black boxes indicate protein kinase C phosphorylation sites. Potential PKA phosphorylation sites were found in PV152-α-AI and PV20-α-AI sequences (blue box). The amino acid residues Y209, S212, and Y213 of the β subunit, important for interaction with the α-amylase enzyme, were conserved in the PV28-α-AI protein (highlighted in grey). An asterisk (*) denotes identical residues (94%); double dots (:) represent a conserved residue substitution; a single dot (.) shows partial conservation of the residue. Black boxes indicate conserved regions.

**Figure 5 plants-12-02918-f005:**
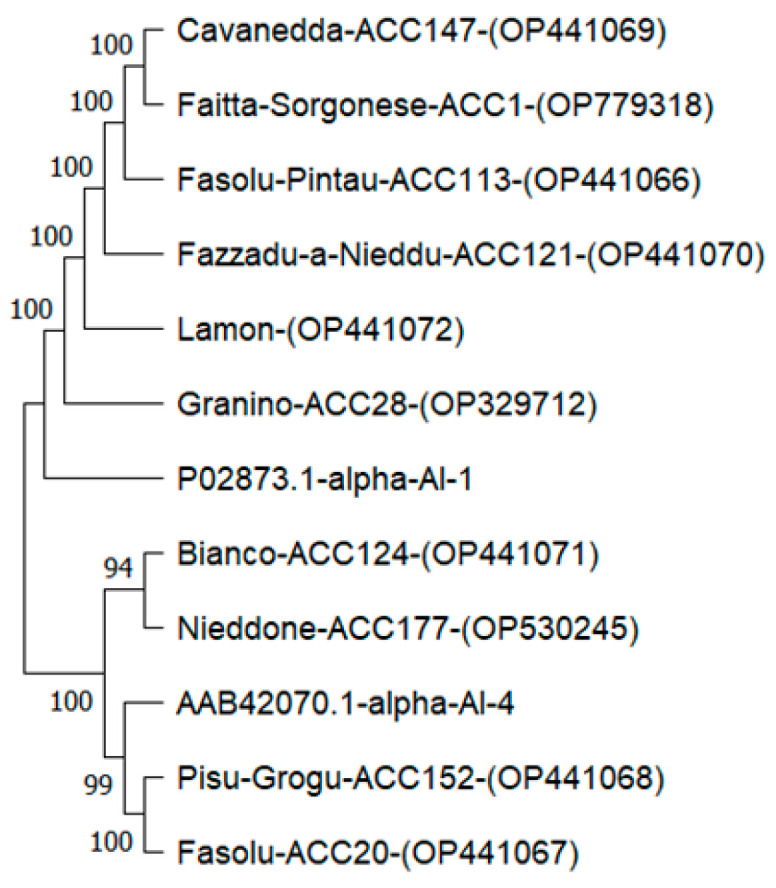
Phylogenetic analysis of α-AI amino acid sequences. The phylogenetic tree was constructed using the Mega 11 tool. Values above branches indicate bootstrap percentages (500 replicates).

**Figure 6 plants-12-02918-f006:**
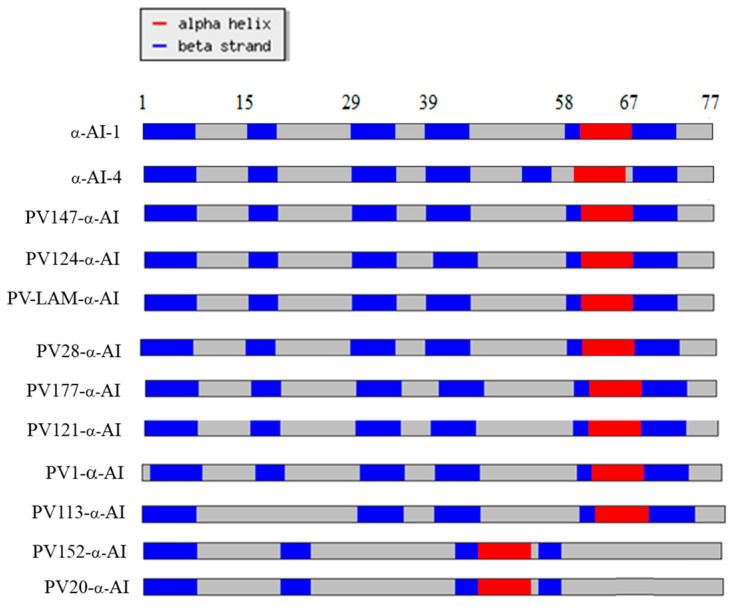
Chou–Fasman (http://cib.cf.ocha.ac.jp/bitool/MIX/, accessed on 21 May 2023) prediction of the secondary structures of the α-AI region that included 77 amino acids of the β-subunit from ten *P. vulgaris* cultivars.

**Figure 7 plants-12-02918-f007:**
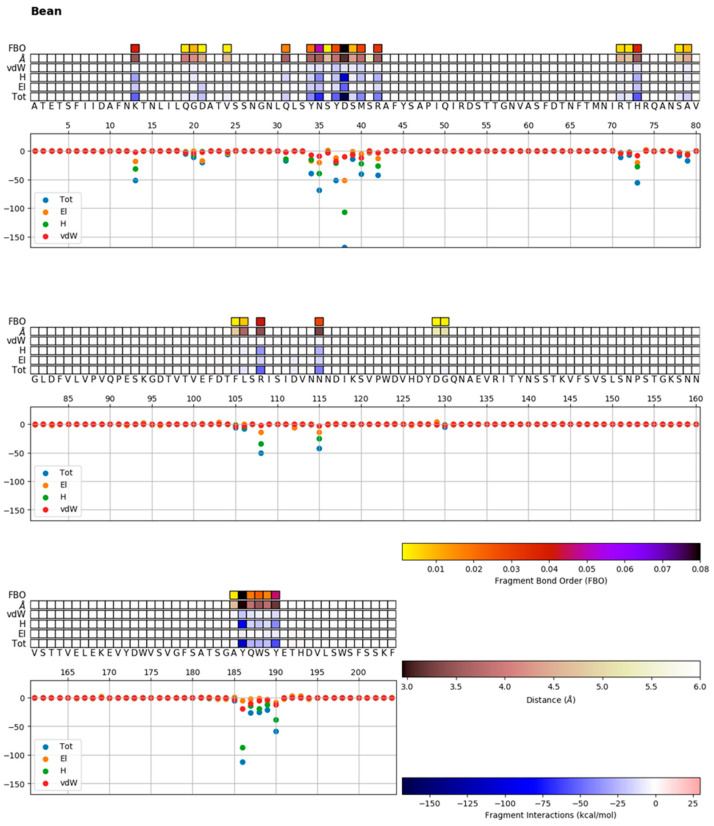
Mechanistic characterization of the binding between α-amylase and its inhibitor from the bean *Phaseolus vulgaris* using the 1DHK PDB porcine pancreatic α-amylase enzyme and *P. vulgaris* α-AI (P02873) amino acid sequence as a model. The abbreviations (Å, Tot, EI, H, vdW) are explained in the text.

**Figure 8 plants-12-02918-f008:**
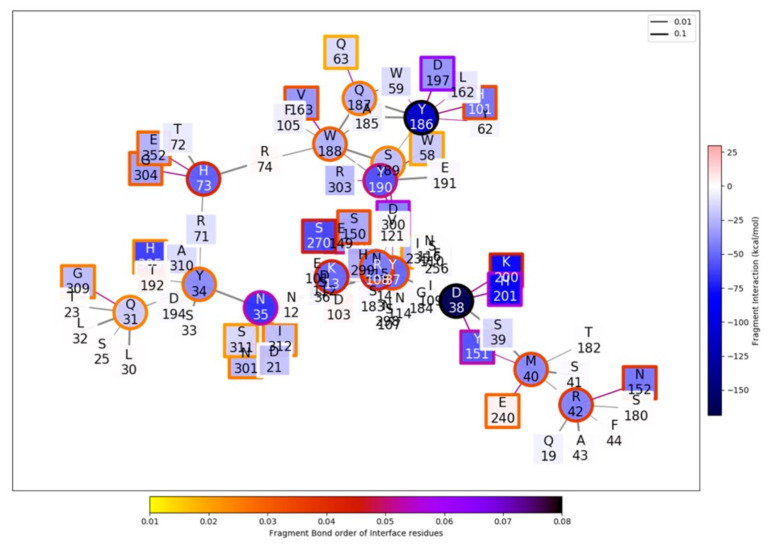
Emerging interaction network that connects the relevant residues of α-AI from common bean with α-amylase (using the 1DHK PDB porcine pancreatic α-amylase enzyme and *P. vulgaris* α-AI (P02873) amino acid sequence as a model).

**Figure 9 plants-12-02918-f009:**
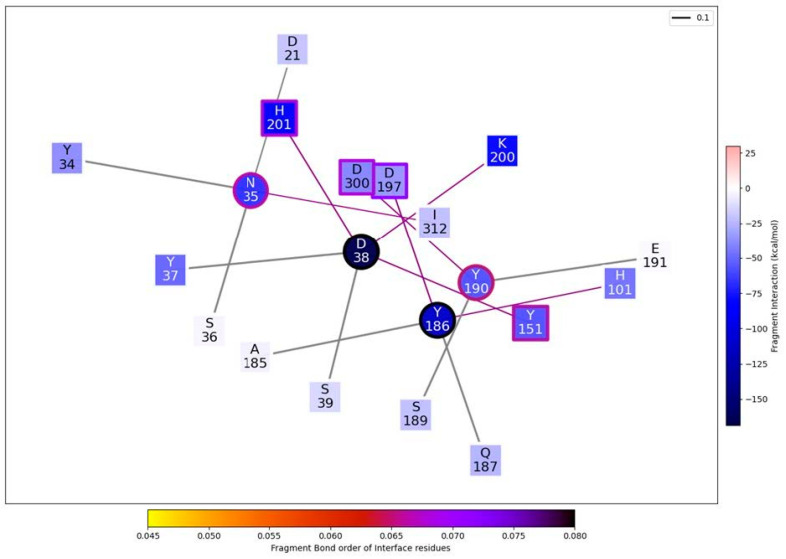
Enlargement of the relevant part of the general interaction graph presented in Figure 9, using the 1DHK PDB porcine pancreatic α-amylase enzyme and *P. vulgaris* α-AI (P02873) amino acid sequence as a model.

**Figure 10 plants-12-02918-f010:**
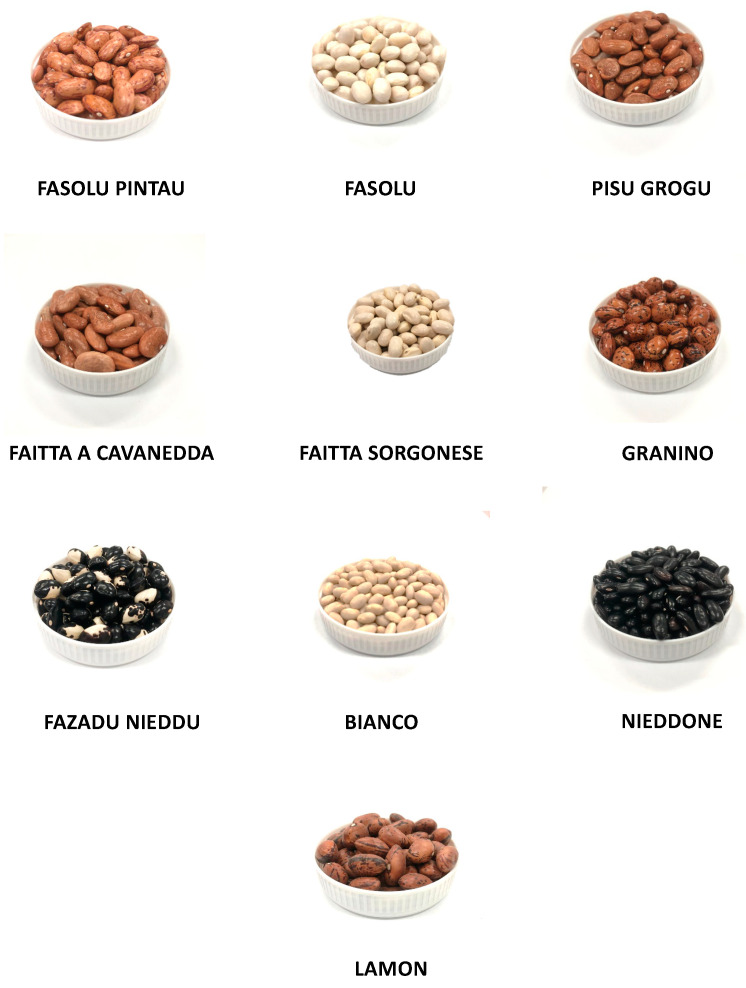
Samples of *P. vulgaris* cultivars.

**Figure 11 plants-12-02918-f011:**
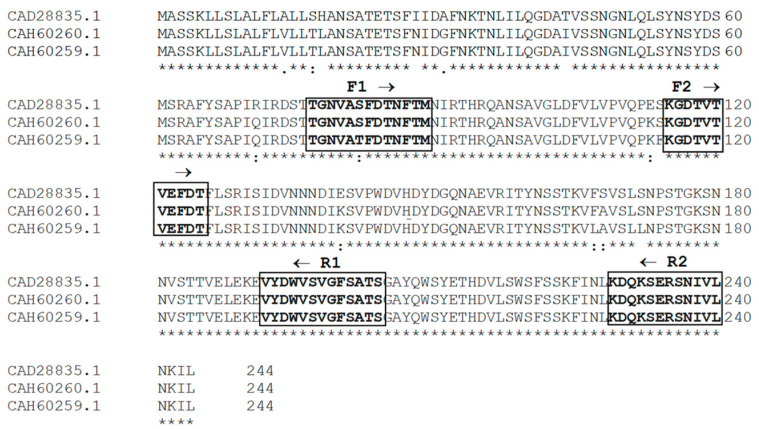
Multiple alignment of α-AI amino acids sequences from three different *Phaseolus* sources. The sequences were chosen from the GenBank SwissProt database and aligned using Clustal Omega (https://www.ebi.ac.uk/Tools/msa/clustalo/, accessed on 6 March 2023). The figure shows the alignments on which the CODEHOP primers were designed. An asterisk (*) denotes identical residues; double dots (:) represent a conserved residue substitution; a single dot (.) shows partial conservation of the residue. The arrow above the amino acid sequences indicates the position of the sense (F →) and antisense (← R) primers chosen from a group of candidate primers obtained from the CODEHOP program. The amino acid sequences chosen by the CODEHOP programme for the design of the primers are in bold inside the boxes.

**Table 1 plants-12-02918-t001:** Inhibitory activity of the examined seed extracts on α-amylase, α-glucosidase, and the total protein content of raw extracts.

Cultivar	Porcine α-Amylase (IAU/g)	Human α-Amylase (IAU/g)	*S. cerevisae* α-Glucosidase (IGU/g)	Total Proteins (mg/g)
Fazadu Nieddu	351 ± 19 ^a^	170 ± 3 ^a^	118 ± 19 ^a,b^	25.8 ± 1.6 ^a^
Bianco	376 ± 12 ^a^	79 ± 6 ^c^	108 ± 18 ^a,b^	35.4 ± 1.7 ^a^
Nieddone	356 ± 26 ^a^	156 ± 6 ^a,b^	66 ± 11 ^b^	33.3 ± 3.8 ^a^
Granino	339 ± 33 ^a^	146 ± 31 ^a,b^	76 ± 8 ^b^	24.2 ± 2.4 ^a^
Lamon	317 ± 17 ^a^	106 ± 6 ^b,c^	97 ± 12 ^a,b^	26.7 ± 1.0 ^a^
Fasolu	287 ± 27 ^a^	196 ± 5 ^a^	182 ± 21 ^a^	22.5 ± 3.0 ^a^
Fasolu Pintau	n.d.	n.d.	76 ± 10 ^a,b^	21.9 ± 2.6 ^a^
Pisu Grogu	320 ± 40 ^a^	115 ± 12 ^b,c^	83 ± 13 ^a,b^	23.2 ± 2.9 ^a^
Faitta Sorgonese	335 ± 30 ^a^	185 ± 30 ^a^	71 ± 22 ^a,b^	24.6 ± 2.9 ^a^
Faitta a Cavanedda	n.d.	n.d.	55 ± 5 ^b^	31.0 ± 5.4 ^a^

Data are expressed as mean ± SEM (n = 5). Mean values for the same analysis having different letters are significantly different (*p* < 0.05; one-way ANOVA followed by the Bonferroni multiple comparisons test). n.d. not detectable. The data are referred to as grams of dry weight.

**Table 2 plants-12-02918-t002:** Haemagglutination activity recorded in 10 common bean cultivars.

Cultivar	MAC (mg/mL)
Fazadu Nieddu	12.5
Bianco	6.25
Nieddone	>200
Granino	12.5
Lamon	12.5
Fasolu	25
Fasolu Pintau	6.25
Pisu Grogu	12.5
Faitta Sorgonese	12.5
Faitta a Cavanedda	12.5

Results are expressed as the minimum protein concentration (MAC, mg/mL) able to agglutinate the sample; n = 5.

**Table 3 plants-12-02918-t003:** *Phaseolus vulgaris* cultivar names (with accession code) and GenBank accession numbers, genes, and protein names corresponding to partial sequences of α-amylase inhibitors.

Cultivar Name and Accession Code	GenBank Accession Number	Gene Name	Protein Name
Faitta Sorgonese (ACC1)	OP779318	pv1-α-AI	PV1-α-AI
Fasolu (ACC20)	OP441067	pv20-α-AI	PV20-α-AI
Granino (ACC28)	OP329712	pv28-α-AI	PV28-α-AI
Fasolu Pintau (ACC113)	OP441066	pv113-α-AI	PV113-α-AI
Fazzadu Nieddu (ACC121)	OP441070	pv121-α-AI	PV121-α-AI
Bianco di Flumini (ACC124)	OP441071	pv124-α-AI	PV124-α-AI
Faitta a Cavanedda (ACC147)	OP441069	pv147-α-AI	PV147-α-AI
Pisu Grogu (ACC152)	OP441068	pv152-α-AI	PV152-α-AI
Nieddone (ACC177)	OP530245	pv177-α-AI	PV177-α-AI
Lamon	OP441072	pv-lam-α-AI	PV-LAM-α-AI

**Table 4 plants-12-02918-t004:** Specificities of *P. vulgaris* cultivars included in this study.

Common Name	Place	Source	Accession Number
Fasolu pintau	Sardinia, Sadali	CBV	ACC. 113
Fasolu	Sardinia, Belvì	CBV	ACC. 20
Pisu Grogo	Sardinia, Austis	LAORE	ACC. 152
Faitta a Cavanedda	Sardinia, Tiana	LAORE	ACC. 147
Faitta Sorgonese	Sardinia, Tiana	CBV	ACC. 1
Granino	Sardinia, Tempio	CBV	ACC. 28
Fazadu Nieddu	Sardinia, Pattada	CBV	ACC. 121
Bianco	Sardinia, Fluminimaggiore	AGRIS	ACC. 124
Nieddone	Sardinia, Ploaghe	AGRIS	ACC. 177
Lamon	Veneto, Belluno	Commercial	BBG-PV.1

**Table 5 plants-12-02918-t005:** CODEHOP oligonucleotides used in PCR experiments and peptides chosen by the CODEHOP programme for the design of the primers.

CODEHOP Sequences	Comments
5′-CGGCAACGTGGCCACCTTCGACACCaayttyacnat-3′	F1 sense primer designed on the conserved peptide TGNVASFDTNFTM
5′-GCTGGTGCCCGTGCAGCCCAAGTCCaarggngayac-3′	F2 sense primer designed on the conserved peptide KGDTVTVEFDT
5′-AGGTGGCGGAGAAGCCCACGGACACccartcrtanac-3′	R1 antisense primer designed on the conserved peptide VYDWVSVGFSATS
5′-TCAGCACGATGTTGGACCGCTCGGAyttytgrtcytt -3′	R2 antisense primer designed on the conserved peptide KDQKSERSNIVL

## Data Availability

All authors agree with MDPI Research Data Policies.

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
