# Peer review of "Biochemical and Phylogenetic Analysis of Italian Phaseolus vulgaris Cultivars as Sources of α-Amylase and α-Glucosidase Inhibitors"

_plants, 2023, doi:10.3390/plants12162918_

Round 1

Reviewer 1 Report

The manuscript presents biochemical and phylogenetic analysis of ten native Italian bean cultivars in the search for their α-amylase and α-glucosidase inhibitor activity. Authors identify also partial cds sequences of α-amylase inhibitor (α-AI) genes from P. vulagris and characterize their predicted protein sequences. In general, the manuscript is well structured and written but some minor points have to be addressed before publication:

Introduction:

Line 39-40: Add „constituting” to “22-25% of seed weight”

Results and Discussion:

Line 145: To what does the statistical significance mentioned here refers? There were no significant differences between Fasolu, Fazadu Nieddu, and Bianco cultivars. The sentence should be corrected.

Line 148: For clarity, “yeast” should be added to “α-glucosidase (p > 0.05; r = 0.31)”.

Line 160: The title of Table 1 should be revised as follows: “of the examined seed extract”

Line 161-162: Does the symbol “g” next to units of activity (IAU/g) in Table 1 refers to grams of fresh/dry weight (FW/DW). If yes, this should be stated in Table 1 or in Material and Methods section.

Line 177: “the lack” should be removed.

Line 207-210: How many repetitions of the haemagglutination activity were performed for the data presented in Table 2? If more than one repetition was performed why are the SD/SE and statistical analysis missing?

Line 238” Why is only the α- subunit of α-AI protein of interest?

Line 242: “synthesized” should be added to “cDNAs”

Line 244: “…allowed us to reconstruct part of the nucleotide sequence of the α-AI gene for each cultivar.” Table 5 from Materials and Methods should be moved to this area of the text since lines 242 – 244 describe the content of Table 5.

Line 244: It should be 62% instead of 70%, according to the analysis of pv28-α-AI sequence from the GenBank database.

Line 245: “pv28-α-AI from Granino cultivar” should replace “pv28-α-AI cultivar” as far as pv28-α-AI refers to the gene name and not to the cultivar name (Table 5).

Line 270: The caption of Figure 1 should be revised as follows: “Alignment of α-AI amino acid sequences from different Phaseolus cultivars with homologous α-AI-1 (P02873.1) and α-AI-4 (AAB42070.1) α-AI isoforms of P. vulgaris from GenBank.

Line 275-276: The caption of Figure 2 should be revised, as already commented for line 270.

Line 307: Should be “N-glycosylation of residues…”

Line 318: The abbreviation “PKA” should be explained at first appearance.

Line 326-327: On Figure 4 the numbers on the left indicating the positions of the amino acids in each protein are not visible.

Line 328: Figure 4 caption should be revised as follows: “Alignment of α-AI amino acid sequences.” Moreover, explanation for the abbreviation, PKA should be given.

Line 336: The sentence should be revised as follows: “Phylogenetic analysis confirmed a notable similarity between the predicted amino acid sequences of PV152-α-AI and PV20-α-AI and the isoform α-AI-4 with respect to isoform α-AI-1, which was closely related to the other eight cultivars.”

Line 341: Figure 5 caption should be revised as follows: “Phylogenetic analysis of α-AI amino acid sequences.”

Line 350: Caption of Figure 6 caption should be revised as follows: “Chou-Fasman (http://cib.cf.ocha.ac.jp/bitool/MIX/) prediction of the secondary structures of the α-AI region that included 77 amino acids of the b-subunit from ten P. vulgaris cultivars.

Line 350: In my opinion, the comparison between secondary structures of α-AI should include α-AI-1 (P02873.1) and α-AI-4 (AAB42070.1) protein sequences since α-AI-1 and α-AI-4 isoforms are included in other bioinformatic analyses (Figures 1-5).

Line 361: “activity” should be added to “α-AI”

Line 377: change “above” for “Materials and Methods section”

Line 390: Is this mechanistic characterization done between porcine pancreatic α-amylase and α-AI (P02873)? If yes, these details should be clearly stated in the text and in the caption of Figure 7. Moreover, reference number for protein sequence of porcine pancreatic α-amylase should be given.

Line 417: In Figure 7 caption, this sentence should be added: ”The abbreviations (, Tot, EI, H, vdW) are explained in the text.”

Line 403: Accession number J01261 refers to the nucleotide sequence. Proper accession number for protein sequence should be given.

Line 404: “amylase” should be revised as “porcine pancreatic amylase”

Line 404: The abbreviation “FBO” should be explained at first appearance.

Line 409: should be “…have not yet been identified and sequenced…”

Line 412-413: The sentence is not clear. To what does the term “no differences” refers? Please clarify. Moreover, the accession number given should be rechecked and corrected (P02873.1).

Line 420: For better clarity, the beginning of sentence should be revised as follows: “On the contrary, for the cultivars other than Granino, it is not…”

Line 424: The description refers to Figure 3A and B not to Figure 5A and B. The same correction of the figure number is required in the proceeding sentence.

Line 438: In the caption of Figure 8, the accession numbers and source organism species names for the stated α-AI and α-amylase should be given.

Line 444: Caption of Figure 8 should be revised as follows: “Enlargement of the relevant part of the general interaction graph presented in Figure 9.”

Materials and Methods:

Line 478: In Figure 10 consider arranging photos of P. vulgaris cultivars in the order corresponding to Table 3.

Line 474: The name of the commercial food supplement should be given.

Line 514: As far as human blood was used, the source of the blood sample and if necessary, a consent of the bioethics committee, should be given.

Line 535: What was the age of plants from which leaves were taken? Why was the seed material not used for RNA extraction?

Line 540: The sentence is not clear. It should be revised as follows: “Total RNA from P. vulgaris for RT‒PCR RNA was extracted from…”

Line 546-549: More details concerning reverse transcription reaction should be given. Was the genomic DNA removed before RT reaction? How much of total RNA was used? How much of each ingredient was added? If RT kit was used, its commercial name and producer should be given.

Line 560: What does the underlined in the third row of the sequence CAH60260.1 in Figure 11 mean?

Line 590: 1 µg of cDNA for RT reaction is a large amount. How was the cDNA concentration measured? Did the Authors probably mean that cDNA was synthetized from 1 µg of total RNA?

Line 591-592: The details of thermal amplification cycles of PCR reaction should be given.

Line 605: This sentence should be moved to Results section.

Line 610: Table 5 should be moved to the Results section. Moreover, the title of Table 5 should be revised as follows: “Phaseolus cultivar names (with accession code) and GenBank accession numbers, genes, and protein names corresponding to partial cds of alpha amylase inhibitors”

Line 633: should be, “Phaseolus cultivar”

Conclusions:

Line 700: replace “some” with “fragments”

Line 708: replace “PO28731” with “P02873.1”

Line 712: Revise as follows, “no inhibitory activity against α-amylase”.

Author Response

The manuscript presents biochemical and phylogenetic analysis of ten native Italian bean cultivars in the search for their α-amylase and α-glucosidase inhibitor activity. Authors identify also partial cds sequences of α-amylase inhibitor (α-AI) genes from P. vulagris and characterize their predicted protein sequences. In general, the manuscript is well structured and written but some minor points have to be addressed before publication:

We thank the Reviewer for the positive comment. We are going to address the specific points.

Introduction:

Line 39-40: Add „constituting” to “22-25% of seed weight”

Done.

Results and Discussion:

Line 145: To what does the statistical significance mentioned here refers? There were no significant differences between Fasolu, Fazadu Nieddu, and Bianco cultivars. The sentence should be corrected.

The Reviewer is right. We amended accordingly the sentence.

Line 148: For clarity, “yeast” should be added to “α-glucosidase (p > 0.05; r = 0.31)”.

Done.

Line 160: The title of Table 1 should be revised as follows: “of the examined seed extract”

Done.

Line 161-162: Does the symbol “g” next to units of activity (IAU/g) in Table 1 refers to grams of fresh/dry weight (FW/DW). If yes, this should be stated in Table 1 or in Material and Methods section.

The data are referred to grams of dry weight. A sentence has been added in the table caption.

Line 177: “the lack” should be removed.

Done.

Line 207-210: How many repetitions of the haemagglutination activity were performed for the data presented in Table 2? If more than one repetition was performed why are the SD/SE and statistical analysis missing?

The repetitions have been at least 5 (this information has been added). However, in almost all cases all the repetitions gave the same concentration as the minimum causing agglutination. So it was not possibile to perform statistical analysis. MAC is a non-continuous variable. So, in this case, a standard deviation value cannot be derived because a significant deviation from the minimum dilution value would have fallen into a higher or lower dilution band.

Line 238” Why is only the α- subunit of α-AI protein of interest?

The interest is for both subunits. “and β”, omitted by mistake, has been added in the sentence.

Line 242: “synthesized” should be added to “cDNAs”

Done.

Line 244: “…allowed us to reconstruct part of the nucleotide sequence of the α-AI gene for each cultivar.” Table 5 from Materials and Methods should be moved to this area of the text since lines 242 – 244 describe the content of Table 5.

Done.

Line 244: It should be 62% instead of 70%, according to the analysis of pv28-α-AI sequence from the GenBank database.

Done.

Line 245: “pv28-α-AI from Granino cultivar” should replace “pv28-α-AI cultivar” as far as pv28-α-AI refers to the gene name and not to the cultivar name (Table 5).

Done.

Line 270: The caption of Figure 1 should be revised as follows: “Alignment of α-AI amino acid sequences from different Phaseolus cultivars with homologous α-AI-1 (P02873.1) and α-AI-4 (AAB42070.1) α-AI isoforms of P. vulgaris from GenBank.

Done.

Line 275-276: The caption of Figure 2 should be revised, as already commented for line 270.

Done.

Line 307: Should be “N-glycosylation of residues…”

Done.

Line 318: The abbreviation “PKA” should be explained at first appearance.

Done.

Line 326-327: On Figure 4 the numbers on the left indicating the positions of the amino acids in each protein are not visible.

We are not sure we understood the numbers that should be more readable. However we tried to increase a little the quality of the image.

Line 328: Figure 4 caption should be revised as follows: “Alignment of α-AI amino acid sequences.” Moreover, explanation for the abbreviation, PKA should be given.

Done.

Line 336: The sentence should be revised as follows: “Phylogenetic analysis confirmed a notable similarity between the predicted amino acid sequences of PV152-α-AI and PV20-α-AI and the isoform α-AI-4 with respect to isoform α-AI-1, which was closely related to the other eight cultivars.”

Done.

Line 341: Figure 5 caption should be revised as follows: “Phylogenetic analysis of α-AI amino acid sequences.”

Done. 

Line 350: Caption of Figure 6 caption should be revised as follows: “Chou-Fasman (http://cib.cf.ocha.ac.jp/bitool/MIX/) prediction of the secondary structures of the α-AI region that included 77 amino acids of the b-subunit from ten P. vulgaris cultivars.

Done.

Line 350: In my opinion, the comparison between secondary structures of α-AI should include α-AI-1 (P02873.1) and α-AI-4 (AAB42070.1) protein sequences since α-AI-1 and α-AI-4 isoforms are included in other bioinformatic analyses (Figures 1-5).

Done. We have prepared a new figure including the secondary structures of isoforms α-AI-1 and α-AI-4.

Line 361: “activity” should be added to “α-AI”

Done.

Line 377: change “above” for “Materials and Methods section”

Done.

Line 390: Is this mechanistic characterization done between porcine pancreatic α-amylase and α-AI (P02873)? If yes, these details should be clearly stated in the text and in the caption of Figure 7. Moreover, reference number for protein sequence of porcine pancreatic α-amylase should be given.

The details have been added in the text (lines 375 and 419).

Line 417: In Figure 7 caption, this sentence should be added: ”The abbreviations (Å, Tot, EI, H, vdW) are explained in the text.”

Added.

Line 403: Accession number J01261 refers to the nucleotide sequence. Proper accession number for protein sequence should be given.

Amended.

Line 404: “amylase” should be revised as “porcine pancreatic amylase”

Done.

Line 404: The abbreviation “FBO” should be explained at first appearance.

Explanation has been inserted.

Line 409: should be “…have not yet been identified and sequenced…”

Done.

Line 412-413: The sentence is not clear. To what does the term “no differences” refers? Please clarify. Moreover, the accession number given should be rechecked and corrected (P02873.1).

The sentence has been rewritten to be more clear.

Line 420: For better clarity, the beginning of sentence should be revised as follows: “On the contrary, for the cultivars other than Granino, it is not…”

Done.

Line 424: The description refers to Figure 3A and B not to Figure 5A and B. The same correction of the figure number is required in the proceeding sentence.

Done.

Line 438: In the caption of Figure 8, the accession numbers and source organism species names for the stated α-AI and α-amylase should be given.

Done. 

Line 444: Caption of Figure 8 should be revised as follows: “Enlargement of the relevant part of the general interaction graph presented in Figure 9.”

Done.

Materials and Methods:

Line 478: In Figure 10 consider arranging photos of P. vulgaris cultivars in the order corresponding to Table 3.

The photos have been re-arranged accordingly.

Line 474: The name of the commercial food supplement should be given.

Added.

Line 514: As far as human blood was used, the source of the blood sample and if necessary, a consent of the bioethics committee, should be given.

Details has been added.

Line 535: What was the age of plants from which leaves were taken? Why was the seed material not used for RNA extraction?

We extracted RNA from approximately 10 day old seedlings. Since we were interested in the determination of the primary structure of the proteins and not in the expression of the mRNA transcribed from the coding gene, we preferred to extract the nucleic acid from the young leaves which contain, compared to the seeds, less starches which imply a lower purification yield. We will consider seed extraction in future RNA quantization studies with Real Time-PCR methods.

Line 540: The sentence is not clear. It should be revised as follows: “Total RNA from P. vulgaris for RT‒PCR RNA was extracted from…”

Corrected.

Line 546-549: More details concerning reverse transcription reaction should be given. Was the genomic DNA removed before RT reaction? How much of total RNA was used? How much of each ingredient was added? If RT kit was used, its commercial name and producer should be given.

The TRI Reagent (Sigma‒Aldrich, St. Louis, MO, USA) used for total RNA extraction involves the destruction of genomic DNA. In lines 559-562 we mistakenly reported the extraction of genomic DNA. We have deleted these lines from the materials and methods of the MS.

More details concerning the reverse transcription reaction have been added in “RNA extraction and reverse transcription” section.

Line 560: What does the underlined in the third row of the sequence CAH60260.1 in Figure 11 mean?

The line was entered by mistake. We have replaced figure 11. 

Line 590: 1 µg of cDNA for RT reaction is a large amount. How was the cDNA concentration measured? Did the Authors probably mean that cDNA was synthetized from 1 µg of total RNA?

The Reviewer is right.  In fact 1 ug is the concentration of RNA used for cDNA synthesis. The cDNA concentration used was 100 ng. We have corrected the concentration of the cDNA used.  

Line 591-592: The details of thermal amplification cycles of PCR reaction should be given.

Done. Thermal cycling steps have been added in the revised MS.

Line 605: This sentence should be moved to Results section.

Done.

Line 610: Table 5 should be moved to the Results section. Moreover, the title of Table 5 should be revised as follows: “Phaseolus cultivar names (with accession code) and GenBank accession numbers, genes, and protein names corresponding to partial cds of alpha amylase inhibitors”

Done.

Line 633: should be, “Phaseolus cultivar”

Done.

Conclusions:

Line 700: replace “some” with “fragments”

Done.

Line 708: replace “PO28731” with “P02873.1”

Done.

Line 712: Revise as follows, “no inhibitory activity against α-amylase”.

Done.

Reviewer 2 Report

The study is well-designed and carried out. I recommend acceptance in the current form. 

Author Response

We thank the Reviewer for the positive comment.

Reviewer 3 Report

It was a very interesting reading. The manuscript is long, but is easy to read.

It shows new results based in new techniques, the conclusions are basically based in your results.

My general  opinion is that it is a good and interesting work that deserves to be published.

Author Response

(The authors gave the same response as above.)
